# THE PRICE OF A SECOND THOUGHT: ON THE EVALUATION OF REASONING EFFICIENCY IN LARGE LANGUAGE MODELS

## ABSTRACT

Recent thinking models trained with reinforcement learning and backward-checking CoT often suffer from overthinking: they produce excessively long outputs even on simple problems, wasting computation. Existing evaluations, based on token efficiency, give an incomplete view as they neglect problem difficulty and intermediate computation costs. We formalize *reasoning efficiency* as a relative measure between thinking and instruct models, treating instruct models as the minimal-effort baseline. A systematic study across four thinking models and multiple benchmarks reveals two consistent patterns: (i) instruct models achieve higher efficiency overall, and (ii) problem difficulty affects efficiency, with thinking models wasting computation on easy problems but providing value on harder ones. Building on this insight, we propose COTHINK, a simple two-stage pipeline: an instruct model drafts a brief outline, and a thinking model expands it. On GSM8K, MATH500, and AIME24, COTHINK cuts token usage by 21.1% while keeping accuracy on four thinking models, and remains competitive with strong efficiency baselines.

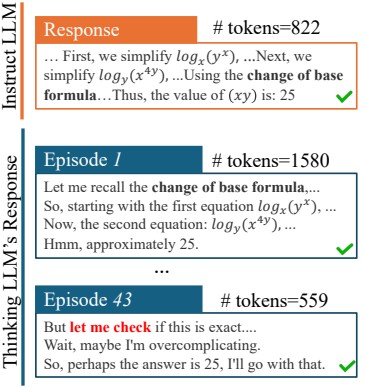

(a) Example output for question Q67

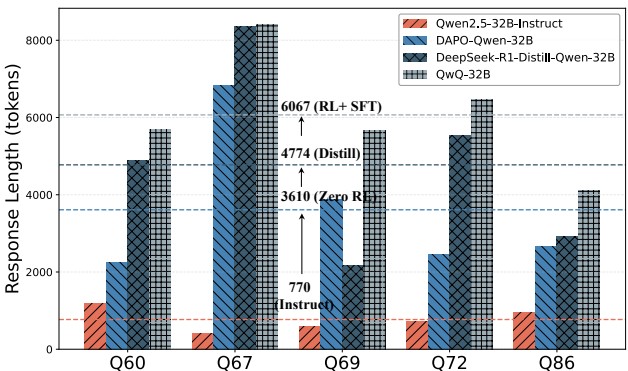

(b) #tokens for 5 questions; dotted lines indicate average

Figure 1: Illustration of token lengths for example questions from AIME 2024, where all models successfully answer all these questions: (a) shows answers by Qwen2.5-32B-Instruct (Instruct LLM) and DeepSeek-R1-distill-Qwen-32B (Thinking LLM) on Q67, (b) plots the total number of tokens in their solutions for 5 questions. Note: Question ID follows the Qwen2.5-Math evaluation format (Yang et al., 2024), ranging from Q60 to Q89.

## 1 INTRODUCTION

Recent thinking models (Jaech et al., 2024; Guo et al., 2025), trained with reinforcement learning (RL) and long chain of thought (CoT) data, outperform non-thinking models on math problem solving (Cao et al., 2025). Unlike general instruction-tuned models, thinking models generate

extended reasoning traces that include multiple rounds of backward-checking CoT wrapped with `think` tags. Following (Qu et al., 2025b), we denote each round of verification as an *Episode*.[1]

Language models trained for complex reasoning often exhibit *overthinking* problem (Chen et al., 2024; Sui et al., 2025), a tendency to generate excessively long outputs that impairs readability and wastes computational resources. On the AIME2024 (university-level mathematics benchmark), for instance, these models produce outputs 5–10 times longer than standard instruction-tuned models of comparable size, even when both solve problems correctly (Figure 1a). This trend is clear in the progressive increase of average output lengths: from 770 tokens for Qwen2.5-32B-Instruct to 3,610 for DAPO (Yu et al., 2025), 4,774 for DeepSeek-R1-Distill (Guo et al., 2025), and 6,067 for QwQ (Qwen Team, 2025b) (Figure 1b).

**Evaluation Limitations.** Prior work has proposed strategies to mitigate overthinking, such as controlling token budgets (Han et al., 2024; Xu et al., 2025), penalizing lengthy responses (Yang et al., 2025b; Luo et al., 2025b), and best-of-$N$ sampling (Fu et al., 2025). Typically, evaluation of reasoning efficiency for a single model is measured by token efficiency (Qu et al., 2025a; Aggarwal et al., 2025), defined as

$$\tau_{(M,D)} = \frac{\mathbf{Q}(D)}{\mathbf{C}_M(D)} \tag{1}$$

where $\mathbf{Q}(D)$ is the quality on dataset $D$ and $\mathbf{C}_M(D)$ is the computational cost of model $M$ on dataset $D$. However, this metric often give an incomplete picture. Firstly, current benchmarks, with their narrow focus on token efficiency in isolated task evaluations, provide a limited and sometimes misleading perspective on model performance. They overlook the critical concepts of *overthinking* and *underthinking* which are relational phenomena observable only through comparative analysis. In complex tasks, for example, a short response that appears efficient may instead indicate *underthinking* and insufficient computational reasoning Aggarwal et al. (2025). Secondly, current benchmarks neglect the costs of intermediate computation, such as ignoring the cost of generating multiple candidate solutions in best-of-N sampling. This focus yields incomplete and biased comparisons, obscuring the principle that total computation should scale with problem difficulty rather than output length alone (Snell et al., 2024; Singhi et al., 2025).

**Relative Efficiency Analysis.** Thus, we consider a more fair evaluation from a relative perspective. By treating the instruction-tuned model as a baseline that reflects minimal reasoning effort, we define the reasoning efficiency of a thinking model relative to this baseline as

$$\eta_{(M_R, M_I)} = \frac{\tau_{(M_R, D)}}{\tau_{(M_I, D)}}, \tag{2}$$

$\eta = 1$ indicates that the reasoning model $M_R$ achieves the same level of efficiency as the instruction-tuned model $M_I$. Values $\eta > 1$ reflect relative gains in reasoning efficiency, while $\eta < 1$ capture efficiency losses. This formulation allows us to quantify not just absolute task performance, but how effectively a model converts additional reasoning into measurable improvements over the baseline.

Under this relative efficiency metric, we evaluate four thinking models with different training algorithms and data distributions against their instruct counterpart across benchmarks of varying difficulty. Our analysis reveals two different patterns. First, instruction-tuned models show higher token efficiency, with most thinking models falling below the baseline. Second, efficiency is strongly difficulty-dependent. Thinking models tend to over-compute and waste computation on simple problems due to long-CoT data patterns, but deliver clear gains on hard problems where instruction-tuned models often falter.

**A Simple Pipeline.** Instruction and thinking language models exhibit complementary strengths. A straightforward strategy is to allocate easy problems to instruct models while engaging deliberate reasoning only for hard cases. In practice, however, even with interfaces like Qwen3 hybrid think mode[2], neither users nor models can reliably assess difficulty in advance. We therefore ask: under

---

[1]No standard criterion exists for segmenting episodes; we use regex patterns like "let me verify" or "on second thought".

[2]This mode allows users to control how much thinking the model performs based on the task at hand.

Table 1: Comparison of general instruct and thinking models in terms of post-training algorithms and Chain-of-Thought (CoT) data strategies (all models use the 32B version).

| Model | Post-train Alg. | | Post-train CoT Data | | | | Focus |
|---|---|---|---|---|---|---|---|
| | SFT | RL | Forward | Backward | Short | Long | |
| Qwen2.5-Instruct (Yang et al., 2024) | ✓ | ✗ | ✓ | ✗ | ✓ | ✗ | General instruct model |
| DAPO (Yu et al., 2025) | ✗ | ✓ | ✗ | ✓ | ✗ | ✓ | RL-only thinking model |
| DPSK-R1-Distill (Guo et al., 2025) | ✓ | ✗ | ✗ | ✓ | ✗ | ✓ | Distillation-based thinking model |
| QwQ Qwen Team (2025b) | ✓ | ✓ | ✗ | ✓ | ✗ | ✓ | SFT+RL thinking model |
| Qwen3 (Qwen Team, 2025a) | ✓ | ✓ | ✓ | ✓ | ✓ | ✓ | Hybrid thinking model |

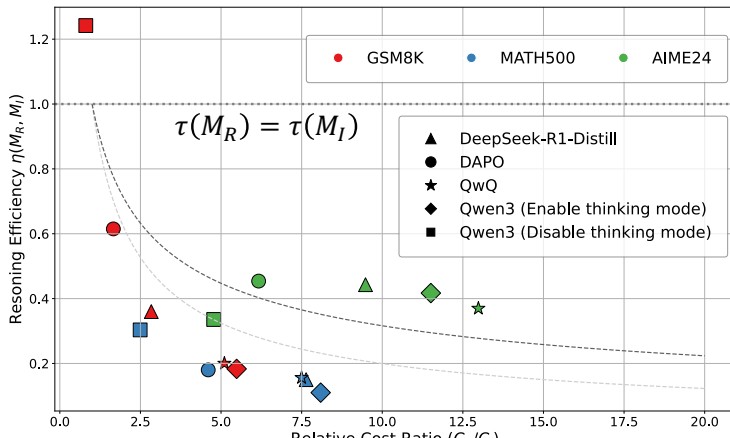

Figure 2: Reasoning efficiency comparison between different model. Each model is represented by a specific marker shape, and each dataset by a distinct color. The dashed gray lines correspond to hypothesized efficiency scaling law with assumed scaling exponents $\beta = 0.3$ and $\beta = 0.5$ for reference.

what conditions do instruct models mitigate overthinking and achieving comparable accuracy with less test-time compute?

Drawing inspiration from sketch prompting (Ning et al., 2023; Beurer-Kellner et al., 2023), we propose **COTHINK**, a simple yet effective two-stage pipeline for efficient reasoning. In the first stage, an instruct model generates a concise solution outline. In the second stage, a reasoning model expands this outline into a complete derivation when necessary. For straightforward problems, the outline itself often suffices, requiring only minimal elaboration. For more challenging problems, the reasoning model naturally produces full derivations.

Concretely, we employ Qwen2.5-32B-Instruct as the outline generator and pair it with four reasoning-oriented models of the same scale: DAPO, DeepSeek-R1-Distill, QwQ, and Qwen3. Across three benchmarks of increasing difficulty: GSM8K (Cobbe et al., 2021b), MATH500 (Hendrycks et al., 2021), and AIME24. COTHINK reduces average computation budget by 21.1% while improving average accuracy by 1.66%.

## 2 REASONING EFFICIENCY: A RELATIVE PERSPECTIVE

We define relative reasoning efficiency in Equation 2 to compare thinking models with their instruct counterparts. In this section, we propose a hypothesized efficiency scaling law, validate it empirically, and analyze how post-training strategies shape inference patterns and their broader implications.

### 2.1 HYPOTHESIZED SCALING LAW FOR REASONING EFFICIENCY

**Experiment Setup.** We evaluate five representative 32B models on three math benchmarks with increasing release time and difficulty (GSM8K, MATH500, and AIME24): one general-purpose

instruct model (Qwen2.5-32B-Instruct) and four thinking models post-trained with distinct algorithms and CoT data Table 1. Together, these models cover various distinct supervision strategies, enabling us to isolate the effects of different training data and algorithms on performance. Notably, Qwen3 is a hybrid thinking model supporting both direct or thinking reasoning, with switchable "thinking" and "non-thinking" modes. Thus, we report Qwen3 results for both settings. To ensure fairness and reproducibility, we use HuggingFace's official `Math-Verify` [3] to validate all generated answers.

**Connection to Test-Time Scaling Law.** Language model performance typically follows a test-time scaling law (Snell et al., 2024; OpenAI, 2024), where response quality improves sub-linearly with increased cost: $Q(C) \propto C^\beta$, with $\beta < 1$. This reflects a phenomenon of diminishing returns—achieving linear gains in quality requires exponential increases in cost. Under this assumption, our reasoning efficiency metric can be approximated as:

$$\eta \approx \left(\frac{C_R}{C_I}\right)^\beta \tag{3}$$

This formulation predicts that as thinking models consume more tokens relative to instruct models, their efficiency advantage should follow a predictable scaling pattern governed by the underlying scaling law exponent. Based on above reasoning efficiency framework, we plot the efficiency metrics for four thinking models across three benchmarks (Figure 2) and derive the following key observations.

**Observation 1 (Instruct Model Shows High Token Efficiency)** *Instruct models produce significantly shorter responses than thinking models, especially on correctly solved questions.*

In Figure 2, the line $\eta = 1$ represents equal reasoning efficiency between thinking and instruct models. Points above indicate superior efficiency over the instruct baseline. Only Qwen3 with thinking mode disabled exceeds the instruct model on GSM8K. All other thinking models fall below $\eta = 1$, showing weaker token efficiency than instruct models. Token efficiency ranking: DAPO > DeepSeek-R1-Distill > QwQ > Qwen3 (Enable thinking mode).

**Observation 2 (Problem Difficulty Affects Reasoning Efficiency)** *Thinking models are more efficient on complex tasks, showing wasted computation on simpler ones.*

Except on the harder benchmark (AIME24), most thinking models remain below the hypothesized efficiency scaling law line, indicating their computational overhead does not yield proportional quality gains. Simple problems trigger overthinking, consuming excessive tokens relative to instruct models. Complex tasks better utilize thinking models' backward checking capabilities, particularly when instruct models struggle or fail entirely.

## 2.2 MECHANISTIC ANALYSIS: SOURCES OF INEFFICIENCY.

For the above two observations, based on the data in Table 1, we identify two key sources of inefficiency in thinking models:

**Algorithmic-level inefficiency.** RL training may unintentionally reduce per-step information density in an episode, encouraging more verbose generation. As Figure 1a shows, thinking models use nearly twice the tokens (1580 *vs* 822) despite following similar logical steps. This observation aligns with prior work Yue et al. (2025), suggesting that RL can promote verbosity. Distillation on data generated by RL models may further amplify this tendency, as seen in models such as DeepSeek-R1-Distill and QwQ.

**Data distribution inefficiency.** Backward CoT training produces multi-episode verification patterns that persist during inference. As Table 1 shows, post-training CoT data include forward-only, backward, long, and short types, reflecting this distribution. Following pattern-matching principles (Vapnik, 2013; Bishop & Nasrabadi, 2006), thinking models tend to repeat checks across episodes even on simple problems, contributing to systematic overthinking.

---

[3]https://github.com/huggingface/Math-Verify

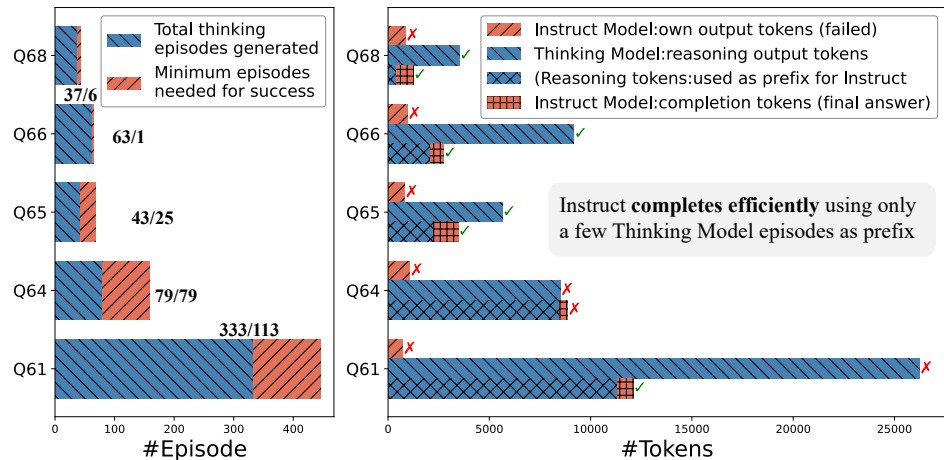

Figure 3: We present five AIME24 questions that the instruct model (Qwen2.5-32B-Instruct) fails to answer on its own. For each question, we prepend thinking episodes generated by the DeepSeek-R1-Distill-Qwen-32B model as context, and test whether this helps the instruct model arrive at the correct answer.

In summary, thinking models overcompute on simple tasks, reducing token efficiency, but provide benefits on complex problems where backward checking is useful. This pattern roughly follows a hypothesized scaling law, with diminishing returns as computation increases. Key sources of inefficiency include RL-induced verbosity and backward CoT data, which together encourage repeated verification even when unnecessary.

## 3 COTHINK

Through reasoning efficiency analysis, instruct and thinking models have complementary strengths. At first glance, a natural solution is to delegate easy tasks to instruct models and reserve harder ones for thinking models. Recent efforts such as hybrid reasoning (Qwen Team, 2025a; Ma et al., 2025; Jiang et al., 2025b; Liu et al., 2025; Luo et al., 2025a; Zhang et al., 2025b) aim to solve this adaptively. For example, Qwen3 (Qwen Team, 2025a) and NoThinking (Ma et al., 2025) implement hard-coded strategies that switch model behavior based on perceived input difficulty.

The fundamental difficulty lies in identifying problem difficulty before solving. Users, and models alike, typically cannot tell how hard a question is until they begin working on it. During the prefill phase, LLMs treat all inputs equally, lacking the means to adapt their reasoning strategy. In practice, difficulty is not a static property of the input—it emerges dynamically during generation. Some problems are solved in a few steps; others require extended reasoning and self-correction. This makes preemptive difficulty assessment inherently unreliable. Prior work often resorts to handcrafted difficulty labels or controlled settings.

### 3.1 CASE STUDY ON AIME24

We compare Qwen2.5-Instruct and DeepSeek-R1-Distill results on AIME24. Outputs fall into three categories: (i) both models solve 5 questions, with the instruct model concise while the thinking model adds verbose steps and backward checks; (ii) on 16 questions, only the thinking model succeeds by correcting errors through verification, which the instruct model cannot perform; (iii) on 9 questions, both fail, with the thinking model attempting longer but still unsuccessful reasoning. Notably, there are no cases where the instruct model succeeds but the thinking model fails.

**Efficient Forward Completion in Instruct Models.** We investigate whether the multiple episodes generated by thinking models are truly necessary. Using the first five AIME24 questions the instruct model fails (Q61, Q64, Q65, Q66, Q68), we prepend DeepSeek-R1-Distill's reasoning episodes as context. Results (Figure 3) show the instruct model solves them with only 27.5% of the episodes and

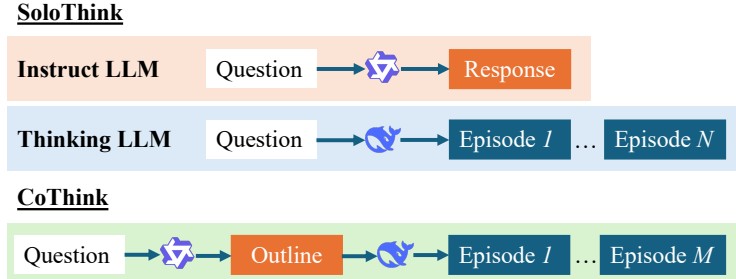

Figure 4: An illustration of the CoThink two-stage framework compared with its SoloThink counterparts using either an instruct model or a thinking model.

11.9% of the tokens compared to the thinking model. This suggests instruct models already possess the necessary knowledge but lack a verification mechanism; when given minimal reflective context, they solve problems more efficiently than thinking models.

**Implications for Reverse Design.** While instruct models efficiently complete reasoning given appropriate episode as prefix, the challenge is predicting the correct number of thinking episodes. This motivates a reverse design: instead of thinking model first, what if the instruct model provides initial guidance? This eliminates episode prediction while leveraging each model's strengths optimally. We propose CoThink, a collaborative pipeline implementing this reverse approach.

## 3.2 PIPELINE

In this context, we propose CoThink, a dynamic two-stage solution process, as illustrated in Figure 4.

**Stage 1: Outline Generation by Instruct Model** The instruct model generates a concise outline without solving the problem, leveraging its high token efficiency in straightforward reasoning to assist the thinking model. The prompts are detailed below:

---

**System Prompt:**

You are a reasoning strategist.
Your job is to break down a complex problem into 2–4 high-level reasoning steps.
Focus only on outlining the general approach or strategy.
Do not include any numbers, formulas, or final answers.
Avoid specific calculations or details—only describe the logic behind solving the problem.

**User Prompt:**

Please break down the following problem.

Problem: {problem}

---

**Stage 2: Backward Verification by Thinking Model** By following the high-density outline from the instruct model, the thinking model efficiently verifies and completes it using fewer tokens.

---

**User Prompt:**

Use only the following steps to solve the problem. Do not change or add steps. Show the work for each step briefly, and place the final answer in \\boxed{}.
Problem: {problem}
Steps: {*outline generated by instruct model*}

---

Interestingly, CoThink *is a more intuitive setup*. For simple tasks, the instruct model's outline is often correct or nearly correct, allowing the thinking model to converge quickly with minimal effort. For harder tasks, the outline provides a structured starting point, enabling the thinking model to apply backward checking and ensure correctness, avoiding unstructured trial-and-error from scratch. This design fundamentally addresses the difficulty assessment challenge: instead of requiring upfront difficulty prediction, the thinking model can dynamically adjust its verification effort based on the outline's quality and correctness.

Table 2: Evaluation benchmarks and average outline tokens produced by the instruct model.

| Dataset | Level | #Samples | #Tokens in ground truth solutions | #Avg outline tokens |
|---------|-------|----------|-----------------------------------|---------------------|
| GSM8K | Primary | 1,319 | [48, 1,070] | 78 |
| MATH50 | High school | 500 | [45, 3,360] | 154 |
| AIME24 | University | 30 | [284, 4,010] | 264 |

## 4 Experimental Validation of CoThink

We evaluate CoThink using the same set of LLMs as in section 2, see Table 1. They include one instruct model Qwen2.5-32B-Instruct and four thinking models trained with different algorithms and CoT data. These models cover diverse supervision strategies, allowing a comprehensive evaluation of CoThink's performance.

### 4.1 Experimental Setup

We evaluate on three math benchmarks, GSM8K, MATH500, and AIME24, covering increasing difficulty and chronological release (Cao et al., 2025), as in Table 2. GSM8K consists of relatively simple grade-school level problems with short solutions, while MATH500 includes more complex high school competition problems. AIME24 is the most challenging, featuring problems from prestigious high school mathematics competitions with significantly longer solutions. This setup helps us validate the insight from the case study, that is whether the instruct model can effectively guide the thinking model to perform token-efficient inference when handling different kinds of questions.

**Baseline.** We compare CoThink against three methods where models solve problems independently, without external outline guidance: (1) *Solo-Thinking*: A single model solves the problem step by step; this reflects typical usage in practical settings. We evaluate both instruct and thinking models in this context. (2) *No-Thinking* (Ma et al., 2025): The thinking model assistant side is prompted with "Okay, I think I have finished thinking." to skip the thinking process and generate the answer directly. (3) *Best-of-N Sampling*: The model generates multiple candidate solutions ($N = 5$), and the shortest one is selected as the final solution.

**Evaluation Metrics.** We evaluate models on both accuracy and computational efficiency. For accuracy, we use **Pass@1**, the percentage of problems solved correctly on the first attempt. For efficiency, we measure **#Tokens**, the total tokens generated per problem, including intermediate generations (e.g., multiple candidates in best-of-N sampling, outline tokens in CoThink). We then compute **Token Efficiency** $\tau$ and **Reasoning Efficiency** $\eta$, as defined in Equation 1 and Equation 2. **Win Rate** is defined as the proportion of evaluation points (across datasets × metrics) where a method demonstrates superiority. A strict win is assigned a score of 1, while a tie is assigned 0.5. The final win rate is computed as the sum of these scores divided by the total number of evaluation points (× Pass@1 × #Tokens × $\tau$ × $\eta$).

### 4.2 CoThink against Baselines

We evaluate CoThink against three baselines: Solo-Thinking, Best-of-N sampling, and No-Thinking, across five thinking models and three math reasoning benchmarks: GSM8K, MATH500, AIME24. The Win Rate analysis in Table 3 provides a comprehensive view of our method's effectiveness across different problem complexities. Our approach demonstrates particularly strong performance on MATH500 (87.5% win rate), indicating high effectiveness on high-school level mathematical reasoning tasks. The method achieves moderate success on university-level problems (AIME24: 60%) and shows room for improvement on elementary problems (GSM8K: 37.5%). Overall, our method attains the best performance in 37 out of 60 evaluation points, resulting in a 61.7% win rate across all model-dataset combinations.

Notably, compared to each model's own Solo-Thinking, on average, CoThink reduces total token usage by 21.1%, reaching up to 41.8% in some cases, while achieving an overall average accuracy

Table 3: Accuracy and efficiency of different reasoning methods across three datasets. The instruct model serves as the baseline reference for reasoning efficiency $\eta$. For CoThink in each setting, improvements over Solo-Thinking of the thinking model are marked in green, declines in red.

| Method | GSM8K | | | | MATH500 | | | | AIME24 | | | |
|---|---|---|---|---|---|---|---|---|---|---|---|---|
| | Pass@1 (%)↑ | #Tokens↓ | τ↑ | η↑ | Pass@1 (%)↑ | #Tokens↓ | τ↑ | η↑ | Pass@1 (%)↑ | #Tokens↓ | τ↑ | η↑ |
| *Instruct model: Qwen2.5-32B-Instruct (as a reference)* | | | | | | | | | | | | |
| - | 96 | 309 | 31.07 | 100 | 82 | 505 | 16.24 | 100 | 16.7 | 1,077 | 1.56 | 100 |
| *Thinking model: DAPO-Qwen-32B (zero RL on Qwen2.5-32B)* | | | | | | | | | | | | |
| Solo-Thinking | 98 | **510** | **19.22** | **61.99** | 67 | 2,025 | 3.31 | 20.38 | 46.7 | 6,639 | 0.70 | 45.36 |
| Best-of-N | 98 | 2,611 | 3.75 | 12.11 | 65 | 11,464 | 0.57 | 3.49 | **60** | 30,210 | 0.20 | 12.81 |
| No-Thinking | 98 | 516 | 18.99 | 61.27 | 68 | 2,742 | 2.48 | 15.27 | 46.7 | 6,965 | 0.67 | 43.24 |
| CoThink | +0.0% 98 | +6.3% 542 | 18.08 | 58.33 | +35.8% **91** | -15.7% **1,707** | 5.33 | 32.83 | -14.3% 40 | -29.4% **4,686** | **0.90** | **58.34** |
| *Thinking model: DeepSeek-R1-Distill-Qwen-32B (Distilled from Qwen2.5-32B)* | | | | | | | | | | | | |
| Solo-Thinking | 94.5 | 823 | 11.48 | 37.04 | 94 | 3,199 | 2.94 | 18.10 | 70 | 10,208 | **0.69** | **44.22** |
| Best-of-N | **95.5** | 4,295 | 2.22 | 7.17 | **97** | 15,857 | 0.61 | 3.77 | **76.7** | 57,943 | 0.13 | 8.54 |
| No-Thinking | **95.5** | **449** | **21.27** | **68.61** | 89 | 2,809 | 3.17 | 19.51 | 63.3 | 11,070 | 0.57 | 36.88 |
| CoThink | -2.1% 92.5 | -35.7% 529 | 17.49 | 56.41 | -2.1% 92 | -36.6% **2,027** | 4.54 | 27.94 | -19.0% 56.7 | -12.5% **8,937** | 0.63 | 40.92 |
| *Thinking model: QwQ (SFT + RL on Qwen2.5-32B)* | | | | | | | | | | | | |
| Solo-Thinking | **97.5** | 1,602 | 6.09 | 19.63 | **98** | 3,933 | 2.49 | 15.35 | 80 | 13,977 | 0.57 | 36.91 |
| Best-of-N | **97.5** | 8127 | 1.20 | 3.87 | 97 | 18,887 | 0.51 | 3.16 | 80 | 68,605 | 0.12 | 7.52 |
| No-Thinking | 95 | 1,679 | 5.66 | 18.25 | 96 | 4,047 | 2.37 | 14.61 | 80 | 14,590 | 0.55 | 35.36 |
| CoThink | -3.1% 94.5 | -41.8% **933** | **10.13** | **32.67** | -3.1% 95 | -19.1% **3,183** | 2.98 | 18.38 | +4.1% **83.3** | -16.2% **11,717** | 0.71 | 45.85 |
| *Hybrid thinking model: Qwen3 (Disable thinking mode, SFT + RL on Qwen2.5-32B with mixed long/short CoT)* | | | | | | | | | | | | |
| Solo-Thinking | 96 | **249** | 38.55 | 124.10 | 62 | 1,258 | 4.93 | 30.35 | 26.7 | 5,138 | 0.52 | 33.51 |
| Best-of-N | 96.5 | 1,351 | 7.14 | 22.99 | **64** | 3,891 | 1.64 | 10.13 | **30** | 19,058 | 0.16 | 10.15 |
| No-Thinking | 97 | 266 | **36.47** | 117.38 | 65 | 771 | 8.43 | 51.92 | 33.3 | 4,686 | 0.71 | 45.83 |
| CoThink | -1.6% 94.5 | +25.7% 313 | 30.19 | 97.18 | +4.8% 65 | -41.2% **740** | 8.78 | 54.10 | +24.7% 33.3 | -21% **4,060** | 0.82 | 52.90 |
| *Hybrid thinking model : Qwen3 (Enable thinking mode, SFT + RL on Qwen2.5-32B with mixed long/short CoT)* | | | | | | | | | | | | |
| Solo-Thinking | 96.5 | 1,696 | 5.69 | 18.31 | 73 | 4,085 | 1.79 | 11.01 | 80 | 12,390 | **0.65** | **41.64** |
| Best-of-N | **97** | 8,669 | 1.12 | 3.60 | **74** | 21,454 | 0.34 | 2.12 | **90** | 63,389 | 0.14 | 9.16 |
| No-Thinking | 94.5 | 1,199 | 7.88 | 25.37 | 73 | 3,688 | 1.98 | 12.19 | 73.3 | 12,814 | 0.57 | 36.89 |
| CoThink | -2.1% 94 | -49.9% **850** | 11.06 | 35.60 | 0.0% 73 | -29.2% **2,893** | 2.52 | 15.54 | -8.4% 73.3 | -4.1% **11,888** | 0.62 | 39.76 |
| **CoThink Performance Summary** | | | | | | | | | | | | |

| Win Rate | GSM8K | MATH500 | AIME24 | Average |
|---|---|---|---|---|
| | 7.5/20 (37.5%) | 17.5/20 (87.5%) | 12/20 (60%) | 37/60 (61.7%) |

improvement of 1.66% across all tasks. This efficiency gain is achieved without requiring prior estimation of problem difficulty, making CoThink a practical choice for many scenarios.

CoThink improves efficiency across all three thinking models, with the overall trend being: QwQ > DeepSeek-R1-Distill > DAPO. QwQ and DeepSeek-R1-Distill benefit from SFT training, making them better at following the outline instructions in CoThink. In contrast, DAPO, trained via RL from a base model, follows less consistently. However, in tasks like MATH500, where DAPO performs poorly but the instruct model does well, CoThink shows strong guiding ability. These consistent efficiency gains across diverse thinking models highlight the generality and robustness of CoThink in improving reasoning efficiency. In particular, on the most challenging dataset, AIME24, CoThink benefits from the strongest thinking model, QwQ, to achieve the highest pass@1 accuracy along with the best token and reasoning efficiency, demonstrating its potential to complement models with strong reasoning capabilities.

Figure 5 shows reasoning efficiency $\eta$ changes from Solo-Thinking to CoThink across three models and datasets, with reference curves at $\beta = 0.3$ and $0.5$. When $\eta = 1$, the reasoning model matches the instruct model's token efficiency; all reasoning models fall below this threshold.

Complex tasks show higher reasoning efficiency than simple ones, as simple problems often lead to overthinking while complex tasks better utilize the reasoning model's strengths. The hollow markers representing CoThink consistently show efficiency improvements, validating our approach.

## 5 RELATED WORK

Token efficiency refers to the problem-solving quality achieved per unit of computation (Qu et al., 2025a), capturing the trade-off between performance and cost. Different models show opposite issues: instruct models often underthink on hard tasks, while thinking models tend to overthink, generating redundant steps even for simple ones (Feng et al., 2025; Sui et al., 2025; Chen et al., 2024; Wang et al., 2025b). To mitigate overthinking, some works limit output length via prompts (Han et al., 2024; Xu et al., 2025; Aytes et al., 2025; Yan et al., 2025), encourage early stopping (Zhang et al.,

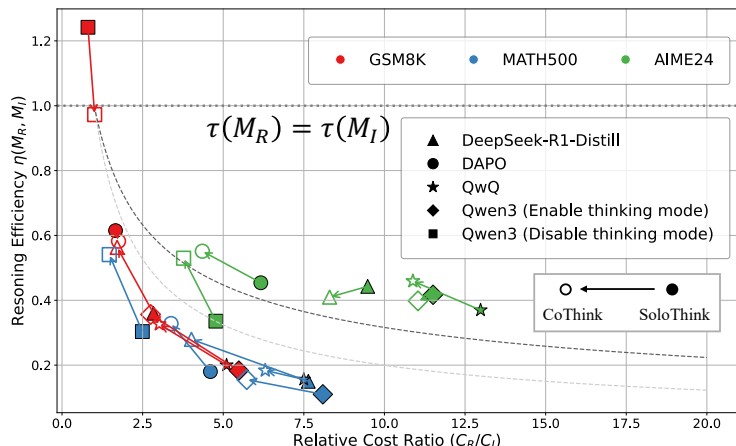

Figure 5: Reasoning efficiency comparison between Solo-Thinking and CoThink.

2025a; Yang et al., 2025a; Jiang et al., 2025a), RL with length penalty (Luo et al., 2025b; Aggarwal & Welleck, 2025; Arora & Zanette, 2025) or SFT on short-solution (Yang et al., 2025b; Xia et al., 2025; Kang et al., 2025). However, these approaches tend to reduce token usage only superficially, and on more challenging tasks, shorter outputs might compromise accuracy by limiting the necessary "thinking time". A more natural way is to assign easy tasks to instruct models and harder ones to thinking models. Recent hybrid reasoning methods (Qwen Team, 2025a; Ma et al., 2025; Jiang et al., 2025b; Liu et al., 2025; Luo et al., 2025a; Zhang et al., 2025b) adaptively assign tasks based on perceived difficulty, for example, Qwen3 (Qwen Team, 2025a) and NoThinking (Ma et al., 2025) use hard-coded switching rules.

A key challenge lies in whether LLMs can perceive problem difficulty. As LLMs are often treated as black boxes, prior work has explored this indirectly through interpretability methods. Some analyze attention patterns show that CoT helps LLMs reason on harder problems (Schnabel et al., 2025; Edelman et al., 2022; Roy et al., 2021). Others disrupt CoT via prompt perturbations (Turpin et al., 2023; Chen et al., 2025), revealing a disconnect between what the model "knows" and what it "says": on simple problems, generation often lacks faithfulness, whereas complex tasks trigger more genuine reasoning. We hypothesize that LLMs treat all inputs equally during the prefill phase, and difficulty emerges dynamically during generation. For example, models may find the correct answer early but keep generating redundant steps due to learned patterns. This dynamic assessment remains underexplored and lacks accurate evidence.

We propose an extremely simple two-stage collaborative pipeline inspired by sketch prompting engineering (Khot et al., 2022; Cobbe et al., 2021a; Beurer-Kellner et al., 2023). Several concurrent works explore related directions. Thought Manipulation (Liu et al., 2025) inserts a pre-generated CoT between the thinking model's `think` tag, allowing the model to better leverage external reasoning. Scot (Wang et al., 2025a) runs a lightweight model in parallel to draft multiple CoT sketches, from which a thinking model selects. In contrast, our method transfers the dense forward reasoning of instruction-tuned models into thinking models via a high-quality outline, requiring no architectural changes and enabling low-cost, deployment-friendly reasoning gains.

## 6 CONCLUSION

In this study, we formalize reasoning efficiency as a relative metric comparing thinking models with instruct counterparts and uncover two key patterns: thinking models are generally less token efficient, and problem difficulty strongly affects computational efficiency. Our results reveal consistent trends across model types and datasets, suggesting the existence of a potential reasoning efficiency scaling law in LLMs. This metric may offer a unified basis for comparing reasoning capabilities across models and datasets. While still speculative, it provides a useful perspective for examining future trade-offs between reasoning accuracy and computational cost.

## ETHICS STATEMENT

This work proposes a method to improve computational efficiency in mathematical reasoning tasks. We only use publicly available datasets (GSM8K, MATH, AIME) and open-source models, without involving human subjects or creating new datasets. The method reduces computational costs in AI reasoning systems, promoting broader access to AI capabilities. We identify no major ethical concerns regarding harmful applications, bias amplification, or privacy violations. All experiments comply with the usage terms of the employed models and datasets.

## REPRODUCIBILITY STATEMENT

We provide detailed implementation to ensure reproducibility. Section 3 describes the pipeline, and Section 4 specifies model configurations and hyperparameters. All datasets are publicly available with standard evaluation protocols. Source code and evaluation scripts are included in the supplementary materials.

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
