# OpenReview forum: "The Price of a Second Thought: On the Evaluation of Reasoning Efficiency in Large Language Models"
_ICLR.cc/2026/Conference — ICLR 2026 Conference Withdrawn Submission_

### Official Review · Reviewer_R1kH · 2025-10-31

**Soundness:** 2
**Presentation:** 3
**Contribution:** 2
**Rating:** 2
**Confidence:** 3

**Summary:**

This paper studies efficient reasoning in large language models (LLMs). The authors point out that reasoning LLMs trained with RL and backward-checking CoT exhibit strong long-form reasoning ability but also tend to suffer from overthinking. Existing efficiency optimization approaches mainly focus on token-level efficiency metrics, which ignore problem difficulty and intermediate reasoning cost, thus failing to distinguish between overthinking and underthinking.

To address this, the authors define a relative reasoning efficiency metric by comparing the token efficiency of a reasoning model against that of an instruct model. They identify two key observations: (i) instruct models are overall more token-efficient; and (ii) reasoning models show advantages mainly on hard problems.

Based on these observations, the authors propose a two-stage pipeline called COTHINK: an instruct model first generates an outline, and a reasoning model then expands it. Experiments on GSM8K, MATH500, and AIME24 show that COTHINK improves accuracy and reduces computation budget, demonstrating its effectiveness.

**Strengths:**

1.	The paper is overall motivative. Through visual analyses such as Figure 1 and Figure 2, the paper illustrates the overthinking phenomenon and its strong correlation with task difficulty, providing a well-motivated foundation for the proposed efficiency metric.
2.	The proposed method is reasonable. It proposes a two-stage pipeline, COTHINK, which uses an instruct model to draft a brief outline, and a thinking model to expand it.
3.	Experimental results demonstrate the effectiveness of the proposed method.

**Weaknesses:**

1. The findings for motivation is similar in existing studies. The two main observations in Section 2.1 (that instruct models are more efficient and reasoning models mainly help on hard problems) have already been reported in multiple prior works (e.g., AutoThink [1], Chen et al. [2], Sui et al. [3], Wang et al. [4]). These studies also show similar reasoning efficiency distributions across problem difficulty and input length.
2. The novelty is somewhat limited. Similar pipeline-based approaches already exist in the same direction (e.g., LM-guided CoT [5]) that also combine small and large models for outline generation and reasoning expansion. The contribution of COTHINK is therefore incremental. Besides, the generated outline can also contain errors, which may lead to potential cascading errors.
3. The differences between COTHINK and prior methods such as Sketch-of-Thought [6], FlashThink [7] are not clearly articulated. These works are mentioned in the related work section but without explicit comparison or further discussion to show advantages.


**REFERENCES**

[1] Tu S, Lin J, Zhang Q, et al. Learning When to Think: Shaping Adaptive Reasoning in R1-Style Models via Multi-Stage RL[J]. arXiv preprint arXiv:2505.10832, 2025.  NeurIPS 2025

[2] Chen X, Xu J, Liang T, et al. Do not think that much for 2+ 3=? on the overthinking of o1-like llms[J]. arXiv preprint arXiv:2412.21187, 2024.

[3] Sui Y, Chuang Y N, Wang G, et al. Stop overthinking: A survey on efficient reasoning for large language models[J]. arXiv preprint arXiv:2503.16419, 2025.  TMLR 2025


[4] Wang Y, Liu Q, Xu J, et al. Thoughts are all over the place: On the underthinking of o1-like llms[J]. arXiv preprint arXiv:2501.18585, 2025.

[5] Lee J, Yang F, Tran T, et al. Can small language models help large language models reason better?: LM-guided chain-of-thought[J]. arXiv preprint arXiv:2404.03414, 2024. COLING 2024

[6] Aytes S A, Baek J, Hwang S J. Sketch-of-thought: Efficient llm reasoning with adaptive cognitive-inspired sketching[J]. arXiv preprint arXiv:2503.05179, 2025.

[7] Jiang G, Quan G, Ding Z, et al. Flashthink: An early exit method for efficient reasoning[J]. arXiv preprint arXiv:2505.13949, 2025.

**Questions:**

Please see strengths and weaknesses.

Besides, it would be helpful to include more discussion or ablation studies against existing training-free reasoning-efficiency approaches to highlight the unique contribution of COTHINK.

In addition, it would also be beneficial to clarify how COTHINK differs conceptually and technically from concurrent two-stage reasoning frameworks, such as related directions’ work: Thought Manipulation and Scot.

---

### Official Review · Reviewer_cX2J · 2025-11-01

**Soundness:** 2
**Presentation:** 2
**Contribution:** 2
**Rating:** 2
**Confidence:** 4

**Summary:**

this paper focuses on reasoning efficiency in thinking models, providing a clean definition of thinking efficiency, analyzing thinking and non-thinking models, and provide a two-stage prompting method to enhance efficiency

**Strengths:**

1. a clean definition for formalization of relative efficiency
2. CoThink works without architectural changes but just prompt engineering with two stages, simple and effective

**Weaknesses:**

1. this topic is also widely and deeply studied, and this paper does not provide new insights or surprising results

2. The mechanistic explanations including RL-induced verbosity and backward CoT patterns are speculative without rigorous evidence

3. Lines 192-194 claim RL reduces "per-step information density" but provide no direct evidence

4. The authors try to establish the scaling law, which has good intention, but how are the parameters fit? The scaling parameters in the Figure are simply drawn for reference, which is not convincing.

**Questions:**

a lot of models now have thing/non-thinking mode switching, in the future would we still need this two stage prompting? is it really necessary?

Other questions see Weakness section

---

### Official Review · Reviewer_2ZwJ · 2025-11-02

**Soundness:** 2
**Presentation:** 3
**Contribution:** 2
**Rating:** 4
**Confidence:** 4

**Summary:**

This paper studies the reasoning efficiency of “thinking” large language models versus standard instruct models. It introduces a Relative Reasoning Efficiency  metric that normalizes compute cost by an instruct baseline, revealing that reasoning models often overthink simple problems but add value for harder ones. Building on this finding, the authors propose a two-stage framework, COTHINK, where the instruct model first generates a short outline and the reasoning model then expands and verifies it. Experiments on three math reasoning benchmarks (GSM8K, MATH500, AIME24) show that COTHINK reduces token usage by about 21% on average with slightly improved accuracy. The paper also explores causes of overthinking and proposes a scaling interpretation for reasoning efficiency.

**Strengths:**

1. The paper introduces an interesting metric that enables consistent comparison across models and tasks. This quantitative perspective fills a gap in evaluating reasoning models beyond simple accuracy or token count.
2. The proposed COTHINK framework is simple and practical. It requires no difficulty prediction, is easy to reproduce, and achieves meaningful compute savings without sacrificing performance.
3. The paper is clearly written and well-organized. Motivation, method, and analysis are coherently connected, and experimental settings are described with sufficient clarity.

**Weaknesses:**

1. While the current experiments focus exclusively on mathematical reasoning, extending the evaluation to at least one non-math reasoning domain (e.g., code generation on HumanEval or MBPP, or knowledge reasoning on GPQA-Diamond or the non-math subset of MMLU-Pro) would strengthen the paper’s generality and demonstrate the broader applicability of the proposed framework.
2. The robustness of the two-stage structure could be explored further. It would be helpful if the author could include at least one ablations that examine (a) the effect of perturbed outlines (such as synonym rewrites, step reordering, small insertions/deletions, or minor errors), and (c) a reversed order setup (reasoning draft → instruct refinement). These analyses would clarify how sensitive the method is to outline quality and the ordering of stages.
3. Given that Table 3 suggests varying accuracy and efficiency patterns across benchmarks, adding a difficulty-level analysis could make the findings more informative. A brief stratified view would quantify the intuition that COTHINK tends to offer greater benefits on harder problems and help identify the most suitable use cases for this approach.
4. Because the proposed strategy-completion paradigm is largely motivated by empirical observations and case studies, including a variant comparison, for example, letting the instruct model produce a complete answer while the reasoning model performs critique and revision (Critique-and-Revise), would further reinforce the empirical validity and credibility of the design.

**Questions:**

See weakness.

---

### Official Review · Reviewer_fcAM · 2025-11-13

**Soundness:** 3
**Presentation:** 3
**Contribution:** 3
**Rating:** 6
**Confidence:** 3

**Summary:**

The authors propose as a fairer way to assess reasoning efficiency in RMs, using a baseline-normalized version of $\tau(M,D)$ (efficiency given dataset D and model M) as $\frac{\tau(M_R,D)}{\tau(M_I,D)}$ (where $M_R$ and $M_I$ are reasoning and instruct models from the same family)

To improve efficiency they propose a two step process where an IM first proposes an outline, and then the RM expands the outline of the CoT into a full one. It improves avg budget by 21.1% and accuracy by 1.66% over GSM8K, MATH500, AIME24.

**Strengths:**

CoThink is simple and delivers performance gains for some models on some tasks. On average, models do gain in performance across some tasks. This suggests that depending on the decision making process, it may be worth trying this approach for some applications.

**Weaknesses:**

The improvement in performance isn’t that clear-cut. When considered alongside the relative simplicity (which may also read as limited novelty) that is either a pro or a con. I am personally inclined to forgive simple methods more for inconsistent performance gains.

Would be nice to see some comparison to or discussion of methods that force early stopping such as [1,2]

Limited motivation of design choices such as prompts. No discussion of how well the models instruction follow/conform to the outlines. Would be nice to see such analysis (see questions)

[1] Fu Y, et al. “Efficiently Scaling LLM Reasoning with Certaindex” NeurIPS 2025 (https://openreview.net/forum?id=nn51ewu5k2), arXiv:2412.20993

[2] Pu, X, et al. “ThoughtTerminator: Benchmarking, Calibrating, and Mitigating Overthinking in Reasoning Models” COLM 2025 (https://openreview.net/forum?id=oHR862dpMC) arXiv: 2504.13367

**Questions:**

Do you have any ablations on prompt? How do you know you picked the right prompts to elicit the desired summarization behavior?

Do you have any analysis of how well models follow the instructions to use the outline provided? Could that explain inconsistent performance? Are some outlines "better" for some tasks than others?

---

### Note · Authors · 2025-12-08

I have read and agree with the venue's withdrawal policy on behalf of myself and my co-authors.